# Association Analysis to Copy Number Variation (CNV) of *Opn4* Gene with Growth Traits of Goats

**DOI:** 10.3390/ani10030441

**Published:** 2020-03-06

**Authors:** LiJuan Li, Peng Yang, ShuYue Shi, ZiJing Zhang, QiaoTing Shi, JiaWei Xu, Hua He, ChuZhao Lei, ErYao Wang, Hong Chen, YongZhen Huang

**Affiliations:** 1Institute of Bijie Test Area, Guizhou University of Engineering Science, Bijie, Guizhou 551700, China; lilijuanpy@163.com (L.L.);; 2College of Animal Science and Technology, Northwest A&F University, Yangling Shaanxi 712100, China; yp00787@163.com (P.Y.);; 3Institute of Animal Husbandry and Veterinary Science, Henan Academy of Agricultural Sciences, Zhengzhou, Henan 45002, China; vincezhang163@163.com (Z.Z.); sqtsw@163.com (Q.S.);; 4College of Veterinary Medicine, Northwest A&F University, Yangling Shaanxi 712100, China

**Keywords:** copy number variation (CNVs), *Opn4* gene, growth traits, Guizhou goat

## Abstract

**Simple Summary:**

Copy number variation is a common genetic polymorphism and is mainly represented by submicroscopic levels of deletion and duplication which are caused by rearrangement of the genome. It is well known that copy number variations of genes are associated with growth traits of livestock. In this study, we detected the correlation between the copy number variation of the *Opn4* gene and growth traits of Guizhou goats. We found that the copy number variation of the *Opn4* gene had a significant influence on the body length and body weight of Guizhou goats. The results may provide preliminary suggestions into Guizhou goat breeding and new insights into the future of CNV as a new promising molecular marker in animal breeding.

**Abstract:**

Extensive research has been carried out regarding the correlation between the growth traits of livestock and genetic polymorphisms, including single nucleotide polymorphisms and copy number variations (CNV). The purpose of this study was to analyze the CNV and its genetic effects of the *Opn4* gene in 284 Guizhou goats (Guizhou black goat: n = 186, Guizhou white goat: n = 98). We used qPCR to detect the CNV of the *Opn4* gene in Guizhou goats, and the classification results were correlated with the corresponding individual growth traits by SPSS software. The results showed that the *Opn4* gene had a superior effect on growth traits with multiple copy variants in Guizhou black goats, and there was a significant correlation between copy number variation sites and body length traits. Contrary to the former conclusion, in Guizhou white goats, individuals with the Normal copy number type showed superior growth traits and copy number variant sites were significantly associated with body weight traits. Therefore, the CNV of the *Opn4* gene can be used as a candidate molecular genetic marker to improve goat growth traits, speeding up the breeding process of goat elite varieties.

## 1. Introduction

In the past, it was thought that the mammalian visual system relied on two photoreceptors to react to light. Studies have shown that biological responses to light are not all derived from visual photoreceptors, and some light will extend to the retinal ganglion cells (ipRGCs) [1]. The melanopsin gene family (also known as *Opn4*) was first identified in Xenopus [2]. Melanopsin, encoded by the *Opn4* gene, is one of the major candidates for mediating nonvisual photoreceptors. The *Opn4* gene regulates the sleep structure by acting on the optic nerve system [3]. Studies have found that white adipose tissue is affected by a blue-sensitive current, small fat droplets, increased fat degradation rate, and changes in hormone secretion, such as adiponectin [4]. This current is produced by the transient receptor potential canonical (TRPC) channel, which is mediated by the opsin produced by the *Opn4* gene [4,5]. As a G protein-coupled receptor, melanopsin is first activated by the activation of phospholipase C and the production of inositol triphosphate (IP3) and diacylglycerol (DAG). Then, the TRPC channel is activated [6]. The TRPC3 channel is expressed in normal and healthy tissues at a low level [7]. A high level of TRPC3 expression has been observed in cells isolated from adipose vascular stroma, indicating the Ca^2+^ channel protein TRPC3 is a determining factor of vascular endothelial growth factor (VEGF) signaling in adipose tissue cells. The above content suggests that the *Opn4* gene can affect fat metabolism and sleep regulation in animals and has an important influence on the growth and development of animals. Based on our previous research, high-throughput sequencing revealed that there was a 2000-bp copy number variation (CNV) in the goat *Opn4* gene, and the CNV marker was located in the copy number variation region of *Opn4* gene candidate region Chr28:4401501-4403500. Based on this, we researched the relation between the *Opn4* gene copy number variation and the growth trait effects of goats.

Copy number variation is a common genetic polymorphism and is mainly represented by submicroscopic levels of deletion and duplication, which are caused by the rearrangement of the genome, generally ranging from 50 bp to 5 Mb in length [8]. Commonly used detection methods fall into two main categories. One is mainly used to detect unknown CNVs in the genome-wide range, including genomic chips and high-throughput metrology techniques, while the other is mainly used for fixed-point detection or verification of known CNVs. Among them, the chip method mainly includes comparative Genomic Hybridization (CGH) [9] and SNP chips [10]. A new technology, single-cell transcriptome resequencing (SCRS), is now available. The single-cell transcriptome study is the basis and starting point for gene function and structure research, analyzing gene expression differences between cells at the single-cell level. Through a new generation of high-throughput sequencing, the information of gene expression in a single cell can be obtained comprehensively and quickly. Almost all transcript sequence information of a cell in a certain state has been widely used in basic research, including the detection of CNV [11]. In the comparative genomic chip, the oligonucleotide probe chip is widely used because of its high precision, high sensitivity, and small sample requirement. In the various methods for detecting known CNVs, qPCR is a widely used technique [12]. The method is simple in operation, high in sensitivity, and fast in speed. A conservative gene is selected in the PCR as an internal reference gene, and then the copy number variation type and the relative copy number of the target gene is determined by the method of 2^−ΔΔCt^ [13].

In this study, we selected 284 Guizhou black goats and white goats from Guizhou Province as experimental samples to study the copy number distribution of the *Opn4* gene in the Guizhou black goat and white goat populations. The Guizhou black goat and Guizhou white goat are both native breeds and bred for meat. We determined the correlation of the results with growth traits, such as body length (BL), body weight (BW), and so on.

## 2. Materials and Methods

### 2.1. Experimental Animals

The protocols used in this study and for the animals were recognized by the Faculty Animal Policy and Welfare Committee of Northwest A&F University (FAPWC-NWAFU, Protocol number, NWAFAC1008).

In order to study the distribution of the *Opn4* gene copy number variation in Guizhou goats, 284 adult goats of the two breeds, Guizhou black goats (n = 186) and Guizhou white goats (n = 98) in the Guizhou Province, were selected. All of the above sample individuals underwent the same feeding management conditions. The growth data of article contained body weight, body length, height, heart girth, and cannon bone circumference. The growth data and the procedure followed for data collection are reported in Table 1.

### 2.2. Sample Collection and DNA Extraction 

Goat ears were used as sampling sites. The ear tissues were collected into 1.5 centrifuge tubes filled with storage liquid (alcohol) and brought back to the laboratory using an ice box. In the laboratory, ear tissues were stored at −80 °C. Next, we weighed 30 mg of the tissue sample, placed it in a 2.0-mL centrifuge tube, cut it into powder with scissors, digested with proteinase K, and extracted genomic DNA using the phenol-chloroform method. The DNA concentration was measured using a spectrophotometer. The dilution volume was calculated by measuring the specific concentration of each sample to ensure each available DNA sample concentration was 10 ng/uL after dilution.

### 2.3. Design and Detection of Primers

In this study, we used the goat *Opn4* gene sequence (NC_030835.1), which was published in the NCBI database, as the reference sequence. The region of the *Opn4* gene CNVs, the chr28: 4401501 bp-4403500 bp, was determined by the results of sequencing. The real-time fluorescent quantitative PCR primer pairs (primer pair P1) were designed by the Primer 5.0 and used to detect the copy number variation of *Opn4* gene in different sample individuals. At the same time, we used the goat *Opn4* gene sequence (NC_030825.1) as the reference sequence, and designed the qPCR primer (primer pair P2) of the *MC1R* gene using the same method (primer pair sequence information is shown in Table 2). When we detected primers, the total volume of the PCR amplification reaction system was 10 µL, including 5 µL 2× Taq PCR Master Mix, 3 µL ddH2O, 0.5 µL per each primer (100 pmol/µL), and 1.0 μL DNA sample pool (10 ng/µL). The DNA sample pool was composed by the genome DNA of 30 samples. The PCR procedure for detecting primers was as follows: Pre-denaturation at 95 °C for 5 min. Then, 35 cycles were conducted: 94 °C for 30 s, 60 °C for 30 s, 72 °C for 30 s, and elongation was conducted for 5 min at 72 °C. Finally, primers were stored at 12 °C.

### 2.4. Statistical Analysis

We amplified each sample by qPCR with primers of target sequence and primers of reference sequence and set three replications per pairs of primer. Subsequently, we analyzed the qPCR results using log2(2−ΔΔCt). The specific calculation method was as follows: ΔΔCt = ΔCt (experimental group) −ΔCt (reference group), ΔCt (experimental group) = Ct (experimental target gene) −Ct (experimental group Internal reference gene), ΔCt (reference group) = Ct (reference group target gene) −Ct (reference group reference gene). Ct is the Cycle threshold, which is the number of amplification cycles. The amplification cycle is the PCR product, which experiences cycling when the fluorescence signal of the amplified product reaches a set threshold. According to the −ΔΔCt method, the quantitative results were divided into three categories: Duplication: −ΔΔCt > 0.5, Deletion: −ΔΔCt < −0.5, and Normal: −0.5 ≤ −ΔΔCt ≤ 0.5. In the study, all analyses were done in two steps. First, a full animal model was used, and then a reduced animal model was used. The full animal model included fixed effects of marker genotype, birth year, season of birth, age of dam, sire, herd, sex, and random effects of the animal. The reduced model was used in the final analysis. The effects associated with herd, sex, sire, age of dam, and season of birth were not matched in the linear model because the preliminary statistical analyses indicated that these effects did not have a significant influence on the variability of traits in the analyzed populations. Thus, we used a fixed effect reduced model. IBM SPSS Statistics 23.0 software was used to analyze the connection between growth traits and CNV types. 

The complete model is as follows:Yijk = μ + A + Gj + Eijk(1)
where Yijk is an individual phenotypic record, μ is the population mean, Ai is the age effect, Gj is the copy number variation type of each point, and Eijk is a random error. The gene expression abundance was derived from the 2−ΔΔCt method, where ΔCt = Ct (target gene) and −Ct (reference genes). There are three types of copy number variations: Duplication (>0.5), Deletion (<−0.5), and Normal (<| ± 0.5|).

## 3. Results

### 3.1. Distribution of Different CNV Types in Goats

As described above, in order to determine the distribution of *Opn4* gene copy number in Guizhou goats, we chose two goat breeds as sample species: Guizhou black goats and Guizhou white goats. According to the log2(2^−ΔΔCt^) values, we divided the CNV types into three classes as Deletion type (<0.5), Normal type (<|±0.5|), and Duplication type (>0.5). Then, we converted the classes to Deletion (0~2), Normal (2), and Duplication (>2). As revealed in Figure 1, in Guizhou black goats, the popular type was the Deletion type, followed by the Normal type and the Duplication type. The Normal type only accounted for one-third of the Duplication type. Meanwhile, in Guizhou white goats, the gap about distribution of three copy number variants was not large. The Duplication type accounted for 35.7%, the Normal type accounted for 33.7%, and the Deletion type accounted for 30.6%.

### 3.2. Correlation Analysis of CNV Type and Goat Growth Traits

In this study, we correlated the *Opn4* gene copy number variation of the above 284 Guizhou goat samples with the corresponding individual growth traits using the one-way ANOVA method. As shown in the Table 3 and Table 4, in the Guizhou black goat group, individuals with the Deletion copy number variation type had a better growth performance, and the variation site was significantly correlated with body length (*p* < 0.05). In the white goat group, the normal-type individuals grew better, and the variant sites were significantly connected with body weight (*p* < 0.05).

## 4. Discussion

Goats are an important animal resource in China. People provide plentiful products, such as goat milk, goatskin, and goat meat. In order to improve the production efficiency and product quality of goats, people focus on the breeding of goats. The detection of SNP and CNV in goat genes and using them for breeding has gained attention. The CNV has been associated with the treatment of certain diseases [14,15]. Currently, researchers are conducting classification studies on the copy number by identifying the difference in the threshold number of times in the qPCR loop. However, there may be a correlation between CNV and SNP [16], and the two techniques can both be considered in association analyses by identifying the SNP markers for common CNV regions [17].

Opsins are a group of retinal-dependent G protein-coupled receptors. Among them, most studies have focused on the Opn1 and Opn2 in the cone cells and the rod cells present in the retinal photoreceptors [18]. To date, most studies on the *Opn4* gene have been related to the circadian rhythm [19]. The *Opn4* gene encodes the opsin, and it is the third protein that transmits light signals except the two photoreceptor cells. This function may have a regulatory effect on the physiological activities of the animal. In particular, the protein responds to blue light and thus produces a specific current which is transmitted to the fat cells, thereby regulating the secretion of hormones, such as adiponectin, and affecting the size of the lipid droplets [19].

In recent years, more and more research has been conducted on the link between CNV and livestock growth, reproduction, and disease [20,21,22]. In this study, we examined the copy number variation of the *Opn4* gene in Guizhou black goats and Guizhou white goats. We found that the CNV of *Opn4* gene was significantly associated with the growth traits of Guizhou black goats and Guizhou white goats. For Guizhou black goats, individuals of Duplication copy number had an advantage in body length, and there was a great difference among the number of the three copy number types. The individuals of Deletion type accounted for 69.9% of the population, the individuals of Normal type accounted for 25.3% of the population, and the Duplication type only accounted for 5.8%. In the Guizhou white goat population, there was no such great difference in distribution. The difference in the number of three copy number types was small, and individuals of Deletion type accounted for 30% of the total, the Normal type accounted for 33.7%, and the Duplication-type individuals accounted for 35.7% of the population. The individuals with the Normal copy number had more body weight than individuals with the other two types, suggesting that Normal-type individuals performed significantly better than the other two individuals in terms of body weight. Whether the *Opn4* gene has an effect on the growth and development of adipose tissue, the function of the gene in animal growth and reproduction, and a series of signal transduction and regulation mechanisms in the body are still unknown, and further research is needed. The Chinese demand for goat-related products is extremely large, especially mountain wool, cashmere, and so on. Goat products have a great impact on the national economy, which has extremely high requirements for the production and growth performance of goats. With the current gene chip technology, it is possible to make early selections of goats and select good individuals to speed up the selection process.

## 5. Conclusions

The results of the association analysis indicated that individuals with multiple copy variants of the *Opn4* gene in Guizhou black goats had superiority in growth traits, and there was a significant correlation between copy number variation sites and body length traits. In Guizhou white goats, individuals with a Normal copy number variation were superior in growth traits, and copy number variation sites were significantly associated with body weight traits. Therefore, the type of *Opn4* gene CNV can be used as a candidate molecular genetic marker to improve goat growth traits and to improve the speed of breeding process of goat elite varieties.

In summary, we examined the copy number variation of the *Opn4* gene in Guizhou black goats and Guizhou white goats, and demonstrated that this degree of variation affects the growth traits in a certain way clearly. This conclusion was confirmed by correlation analysis, suggesting that the *Opn4* gene plays an important role in the process of Guizhou goat growth. Our research demonstrates the effects of Opn4 CNV in Guizhou goats for the first time in order to provide evidence that the CNV of the *Opn4* gene may be a potential factor for growth traits of other livestock species like bovines.

## Figures and Tables

**Figure 1 animals-10-00441-f001:**
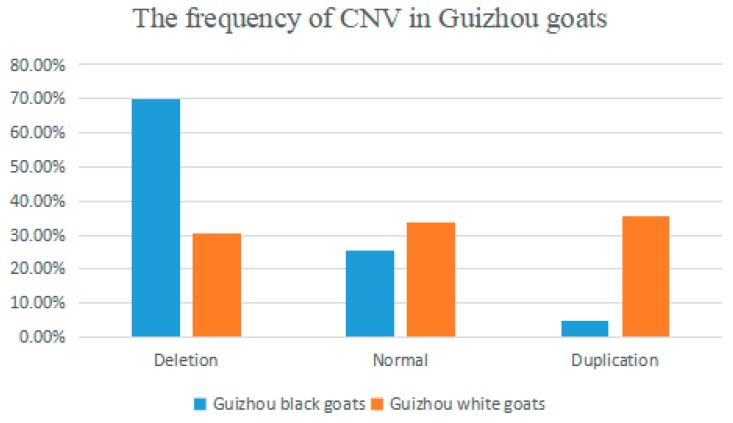
The frequency of CNV in Guizhou goats. Distribution of different copy number variants of *Opn4* gene in Guizhou black goats and Guizhou white goats.

**Table 1 animals-10-00441-t001:** The measured method of growth data.

The Growth Traits	The Measured Method
Heart girth	The circumference of the chest behind the rear edge of scapula.
Withers height	Vertical distance from the highest position of shoulder to the ground.
Body length	The straight distance from shoulder to rear edge of ischial tuberosity.
Body weight	Weighing after stop feeding for a day.

**Table 2 animals-10-00441-t002:** Primer information in copy number variations (CNV).

Primer	Position	Sequence (5’->3’)	Product Length
*Opn4*-CNV	Forward primer	CGTGATACCAGGCTCCAGA	19 nt
	Reverse primer	ACGGCGAGGTTGATAATGA	
*MC1R*	Forward primer	CTCGTTGGCCTCTTCATAGC	21 nt
	Reverse primer	GAAGTTCTTGAAGATGCAGCC	

**Table 3 animals-10-00441-t003:** Correlation analysis between copy number variation of *Opn4* gene and Guizhou black goat growth traits.

Growth Traits (Average Value ± Standard Error)	CNV Types			*p* Value
	Deletion (n = 130)	Normal (n = 47)	Duplication (n = 9)	
BW (kg)	28.113 ± 0.749	28.034 ± 1.245	27.167 ± 2.845	0.95
WH (cm)	59.254 ± 0.538	59.426 ± 0.895	60.000 ± 2.045	0.933
BL (cm)	60.400 ± 0.588 ^a^	62.936 ± 0.978 ^b^	65.222 ± 2.235 ^b^	0.018 *
HG (cm)	74.377 ± 0.608	74.170 ± 1.102	72.222 ± 2.312	0.666

Note: Value with * differs significantly at *p* < 0.05. ^a, b^: Means with different superscripts were significantly different (*p* < 0.05). BW: Body weight. WH: Withers height. BL: Body length. HG: Heart girth.

**Table 4 animals-10-00441-t004:** Correlation analysis between copy number variation of *Opn4* gene and Guizhou white goat growth traits.

Growth Traits (Average Value ± Standard Error)	CNV Types			*p* Value
	Deletion (n = 30)	Normal (n = 33)	Duplication (n = 35)	
BW (kg)	27.133 ± 1.331 ^a^	31.502 ± 1.269 ^b^	26.230 ± 1.232 ^a^	0.010 *
HG (cm)	71.167 ± 1.214	73.773 ± 1.158	70.614 ± 1.124	0.123

>Note: Value with * differs significantly at *p* < 0.05. ^a, b^: Means with different superscripts were significantly different (*p* < 0.05). BW: Body weight. HG: Heart girth.

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
