# Peer review of "Association Analysis to Copy Number Variation (CNV) of Opn4 Gene with Growth Traits of Goats"

_animals, 2020, doi:10.3390/ani10030441_

Round 1
Reviewer 1 Report
Dear Authors,
the manuscript provides some new and interesting results about the relationships between CNV for Opn4 and growth traits. However, the English of the manuscript is really poor. I recommend the Authors to ask to a professional service to edit the English before submitting the revised version. Moreover, it is clear that some mistakes are due to a bad checking of the final version of the manuscript, therefore I recommend to carefully read the paper before any submissions. Furthermore, Discussion should be improved with more considerations about the relationships between Opn4 gene and growth traits. I have also reported some detailed comments above. In some of them I also tried to correct some English mistakes, but the manuscript has much more errors, therefore, again, I recommend a deep English revision. Finally, you should put in the manuscript an Ethic statement, because you used tissue samples.
Simple summary:
L19: “well known” and not “all known”
Abstract:
The first part is too long, too many sentences before arriving at the aim of the study. Moreover, it is not clear why you are studying Opn4 in relation to growth traits, since it encodes opsin, mainly expressed in photoreceptors. Therefore I suggest to remove this part: “CNV is an important component of genomic structural variation. Studies have shown that the Opn4 gene encodes opsin, which is mainly expressed in autologous photoreceptor retinal ganglion cells of the retina. The current research is related to reaction to light and sleep, and there are few studies on livestock growth traits”.
L25: Please, delete “being” or “carried out”
L33: What do you intend with “classification results”? Probably “classification” is not the correct word
Introduction:
L46: … photoreceptors, but some light are extended to…
L47: Please, substitute the comma after [2] with a dot.
L49: “by acting on” instead of “by regulating”
L53: “first” has to be without capital letter
L53: …is activated…
L56: …and indicated…
L62: Do you have a reference for the sentence ending here?
L68: Do you have a reference for the high-throughput metrology techniques? Could you briefly explain what they are?
L72: Do you have a reference for the SCRS?
L73: Please delete one of the two “single”; moreover: a single what??
L78-80: The sentence starting with “A single copy” is not Introduction; please, delete the sentence or move it elsewhere.
L80: 2-ΔΔCt is not correctly written. It has to be 2^-ΔΔCt or 2-ΔΔCt (-ΔΔCt is superscript).
L83: “Including body length (BL), 83 body weight (BW) and so on.” This is not t a sentence. Please, formulate it again
Materials and Methods:
The manuscript lacks of an Ethic Statement about the research on animals. Do you have a document that attests that you research complies with Ethic Standards for Animal Welfare of your country or place?
L87: …in Guizhou goat, 284 adult goats of the two breeds Guizhou black goats (n=186) and Guizhou white goats (n=98) in Guizhou Province were selected.
L89: Could you tell something more about this breed? Is it native, transboundary, crossbred..? Is it bred for milk, meat, wool…?
L90-91: Please, add the verb at the end of the sentence: “Growth data, including body weight, body length, height, Heart girth, and cannon bone circumference were considered”.
L93: It is not explained how did you collect ear samples. You should report this information, in addition to the Ethic statement mentioned above.
L101-104 and L106-109: These sentences are really badly written and totally unclear. There is no subject and verb. Please, rewrite and deeply modify it, also asking the help of a professional service for English editing.
L111: Please put a dot instead of a comma before DNA pool.
L112: …as follows…
L117: We amplified each sample…
L118: …per pairs of primer. Subsequently, we analyzed the qPCR results by…
L119 and L129: Please, write correctly log2(2-ΔΔCt) and put a reference here. You should also report briefly here what this test is doing.
L120: “Deletion” instead of “Deltion type”
L121-122: It is not clear how did you perform the ANOVA. Did you put all the environmental effects in the model? In this case I suggest to call the analysis as “linear model” instead of “ANOVA”. What are the dependent variables of the model? Did you analyze growth traits and CNV using this model? What do you mean with “simplified”? Please, explain.
L122: “Based on the data” is not necessary; please, delete.
Results:
L132: Please, put a dot instead of a comma after “Duplication type”, and then start with a capital letter.
L140-141: This sentence is more proper for Discussion; please, move it.
L141-143: This sentence is more proper for Materials and Methods; please move it. You could start this part of Results like: “Table 2 and Table 3 report the correlations between CNV of Opn4 gene and growth traits respectively in Guizhou black and Guizhou white goats. As shown in the tables, in the Guizhou black group…”
L146: “connected” instead of “connection”
Table 2 and Table 3: Tables should be self-explanatory, therefore you should put in caption the meaning of the abbreviations (BW. WJ, BL and HG). You should also indicate what the * is meaning (I guess it is indicating the significant correlations). Finally, please change “CM” in “cm”.
Discussion:
L153: “Our country” is too colloquial. Please, indicate the name of the country. Moreover, you should also delete “Our” before “people”.
L155: “the detection” instead of “by detecting”
L156-157: Please, change as follows: “In humans CNV has been associated with certain diseases for their treatment [17.18]. At present…”
L157-159: The sentence starting with “At present” is not clear. Could you specify better what do you intend with “the difference in the threshold number of times in the qPCR loop”?
L160: The sentence is unclear; if I have correctly understood what you are saying, please, change it with: “…[19], and the two techniques can be both considered in association analysis by identifying SNP markers for common CNV regions [20)]”
L163: “…the most studied is the Opn1 and Opn2…”
L169: You should put a reference at the end of this sentence.
L172: “Guizhou black goats” at the end of the line has to be removed.
L174-179: These sentences are proper for Results, not for Discussion. Please, move to the results the numbers that have been not already reported there, and delete the rest. You should substitute this part with a more general sentence.
L179-182: Why, in your opinion, normal individuals are better? Could you enlarge the discussion of this part, also introducing new literature and consideration? You also could put some findings from other species for similar traits (if existing).
L185: Please put a dot after “goat” and start a new sentence; “it” has to be written without capital letter. You have a number of similar mistake in the manuscript. Please, check CAREFULLY the revised version before the next submission.
Conclusions:
L190-191: What is normal in white goats? Please, explain better.
L200: “…a potential factor for growth traits of other livestock species like bovines”
Reviewer 2 Report
The Authors analyzed the correlation between growth traits and the frequency of the CNV located on Opn4 gene in Guizhou black goats and in Guizhou white goats. They used qPCR for detecting CNV located on Opn4 gene.
The English language needs to be checked throughout there are multiple minor errors and awkwardly phrased sentences. I strongly encourage the authors to have the manuscript read by a native English speaker. At present, it is in places difficult to understand what the authors are trying to convey. With these views, my decision is to reject this manuscript for publication in Animals.
Comments for the Authors
Material and methods section
Line 78: How were selected the animals? How were collected the biometric information and phenotypes?
Line 93 to 106: English revision and an explanation of these sentences are needed: e.g. why the Authors used MC1R gene; the primers for qPCR were designed on entire Opn4 gene sequence or on CNV inside the gene.
Primer5.0 is a software for primer design or is a web tools.
Line 108 to 116: English revision and an explanation of these sentences are needed.
Results section
Line 119 to 123: in my opinion, these sentences have to be moved to section “Statistical analyses”
Table 2 and 3: please modify the legend in the tables adding an explanation of the abbreviations
Discussion section
The discussion section have to be improved.
Reviewer 3 Report
The authors have revised the manuscript accordingly with exception.
Author Response
Thank you for your comments.
Reviewer 4 Report
Overall a very good paper that gives the first evidence of the use of Opn4 gene as a marker for growth traits in goats. I have a few suggestions for minor edits below.
Line 83. Including body length... should not be a new sentence, remove the full stop.
Line 90. this sentence is incomplete. Perhaps add to the end of the snetce "was collected"
Line 93: I would change this to "goat ear tissue" instead of goat ears, as you do not use the whole ear. Also include how you got the tissue- was it cut out? Hole punch? Roughly what size was collected?
Line 94: What type and concentration of alcohol?
Table 3: Why does it not show data for WH and BL which is does in Table 2 for the black goats?
Line 155-157 these two sentences do not make sense "It has become a hot spot by detecting the SNP 155 and CNV of goat genes and using them for breeding. Now CNV has been associated with certain 156 diseases for the treatment of certain diseases. The disease offers the possibility [17,18].
Line 162-169: You have already said this in the introduction. Either link to your results or delete.
Line 171-172: you say it was significantly associated with the growth traits of Guizhou black goats but wasnt it only significant for body length? Would this have any affect on the amount of meat or quality of meat when the body weights were the same?
Line 172-173: Change sentence to begin with "For Guizhour black goats, duplication copy..."
Round 2
Reviewer 1 Report
Dear Authors,
the manuscript has been revised but it still need an improvement.
In the following, I have reported a list of questions and response that is in your file of “Author’s reply to Reviewer 1”, that in my opinion need to be considered again. There is a sentence to remove from the abstract (Question 2), and a list of answers you provided that should be also put in the manuscript itself, and not only in the response to reviewer’s concerns. This is the way to effectively improve the manuscript. As you can read below, the sentences starting with “Rev:” are my comments to your sentences, also reported. Please, try to make your best to answer also these additional requests.
Question2: Abstract: The first part is too long, too many sentences before arriving at the aim of the study. Moreover, it is not clear why you are studying Opn4 in relation to growth traits, since it encodes opsin, mainly expressed in photoreceptors. Therefore I suggest to remove this part: “CNV is an important component of genomic structural variation. Studies have shown that the Opn4 gene encodes opsin, which is mainly expressed in autologous photoreceptor retinal ganglion cells of the retina. The current research is related to reaction to light and sleep, and there are few studies on livestock growth traits”.
Response2: Thanks for your suggestion.I have remove the content.
Rev: It is not true. This part is still in the revise ms. Please, remove this part from the new version of ms.
L68: Do you have a reference for the high-throughput metrology techniques? Could you briefly explain what they are?
LINE62
This is based on the results of our sequencing and data analysis.
Rev: Ok, could you indicate in the text something as: “Based on our previous research,…”
LINE68
The second-generation sequencing technology uses the idea of sequencing while synthesizing. It can sequence millions of nucleic acid molecules at once to obtain tens of billions of base sequences and solves the shortcomings of the low throughput of the Sanger sequencing method, so it is also called For High-Throughput Sequencing (HTS).
LINE72
Single-cell transcriptome sequencing technology can quantitatively analyze gene expression differences between cells at the single-cell level. With the rapid development of this technology, the advent of high-throughput single-cell transcriptome sequencing can simultaneously sequence thousands of cells, enabling Heterogeneity analysis of population cells is more efficient.
LINE75
Through a new generation of high-throughput sequencing, the information of gene expression in a single cell can be obtained comprehensively and quickly.
Rev: Could you synthetize these explanations you made and briefly add them to the manuscript, or at least to provide references for SCRS?
LINE89
Guizhou black goat and Guizhou white goat are both native breeds and bred for meat.
Rev: Please, add this information to the manuscript
LINE100
“Growth data, including body weight, body length, height, Heart girth, and cannon bone circumference.” was revised to “The growth data of article contain body weight, body length, height, Heart girth, and cannon bone circumference.”
Rev: I can’t find this part in the manuscript. Could you, please, add it to the new version of ms.?
LINE 136 and 147
Subsequently, we analyzed the qPCR results by using log2(2-ΔΔCt).
Guizhou white goats. According to the log2(2-ΔΔCt)
The experimental results are calculated by using the 2-△△Ct method. The specific calculation method is as follows: ΔΔCt = ΔCt (experimental group) -ΔCt (reference group), ΔCt (experimental group) = Ct (experimental target gene) -Ct (experimental group Internal reference gene), ΔCt (reference group) = Ct (reference group target gene) -Ct (reference group reference gene). Ct is the Cycle threshold, which is the number of amplification cycles. Amplification cycle is the PCR product experienced the times of cycling when the fluorescence signal of the amplified product reaches a set threshold. According to the -ΔΔCt method, the quantitative results are divided into three categories: Duplication: -ΔΔCt> 0.5; Deletion: -ΔΔCt <-0.5; Normal: -0.5≤-ΔΔCt≤0.5 .
Rev: I can’t find these parts in the ms. Please, add them.
LINE140:
In the study, all analyses were done in two steps, first using a full animal model and then using a reduced animal model. The full animal model included fixed effects of marker genotype, birth year, season of birth, age of dam, sire, herd, sex and random effects of animal.
The reduced model was used in the final analysis. An effect associated with herd, sex, sire, age of dam and season of birth were not matched in the linear model, as the preliminary statistical analyses indicated that these effect did not have a significant influence on variability of traits in analyzed populations. So we used a fixed effect reduced model.
Rev: You can include this information in the text
LINE 195
ΔΔCt = ΔCt (experimental group) -ΔCt (reference group), ΔCt (experimental group) = Ct (experimental target gene) -Ct (experimental group Internal reference gene), ΔCt (reference group) = Ct (reference group target gene) -Ct (reference group reference gene). Ct is the Cycle threshold, which is the number of amplification cycles. Amplification cycle is the PCR product experienced the times of cycling when the fluorescence signal of the amplified product reaches a set threshold.
According to the -ΔΔCt method, the quantitative results are divided into three categories: Duplication: -ΔΔCt> 0.5; Deltion: -ΔΔCt <-0.5; Normal: -0.5≤-ΔΔCt≤0.5 .
Rev: You could put this explanation at the end of the M&M, before the Results
LINE 219
The reason of “Normal individuals are significantly better than the other two individuals in terms of weight”: we could find that the Normal copy number individual’s body weight is bigger than other two types’, which means the Normal type produce more meat.
Rev: ok, but please, include this explanation also in the text
Reviewer 2 Report
Line 100: replace the sentence “.....All of the above sample individuals selected the same feeding management conditions.” with the sentence in italics “….All the individuals were reared at the same feeding management conditions.”
Line 101-102: As I reported in the first review, is mandatory knowing how were selected the animals and how were collected the biometric information and phenotypes. Please, add a table with the variables measured and the procedure followed for the data collection. Replace the sentence“….The growth data of article contain body weight, body length, height, heart girth, and cannon bone circumference.” with the sentence in italics “….. The growth data and the procedure followed for data collection were reported in Table 1”.
Line 104 to 106: replace the sentences “Taking black goat and white goat ear tissues into 1.5 centrifugal tubes filled with storage liquid bring them back to the laboratory with ice box, and store at -80 ℃. Goat ears were used as sampling sites. After collecting tissue samples, they were stored in alcohol and stored in a -80℃ refrigerator for laboratory.” with the sentence in italics “Goat ears were used as sampling sites. The ear tissues were collected into 1.5 centrifuge tubes filled with storage liquid (alcohol) and bring back to the laboratory with ice box. In laboratory, they were stored at -80 ℃.”
Line 115 to 132: The English language needs to be checked. The explanation of the laboratory procedure is confused.
Line 165 to 172: modify the sentences “As shown in the table2-3, in the Guizhou black goat group, individuals with a copy number variation type of deletion type has a better growth performance, and the variation site was significantly correlated with body length (P<0.05). In the white goat group, the normal type Iindividuals grow better, and the variant sites were significantly connectioned with body weight (P < 0.05).” with the sentences in italics “As shown in the table 2 and 3, in the Guizhou black goat group, individuals with a CNV deletion type has a better growth performance, and the variation site was significantly correlated with body length (P<0.05). In the white goat group, the normal type individuals have better growth performances, and the variant sites were significantly correlated with body weight (P < 0.05).”
Discussion section
Despite the discussion section is well improved, it still needs an English language accurately revision.
